# Prediction of Epilepsy Based on Tensor Decomposition and Functional Brain Network

**DOI:** 10.3390/brainsci11081066

**Published:** 2021-08-13

**Authors:** Han Li, Qizhong Zhang, Ziying Lin, Farong Gao

**Affiliations:** Institute of Intelligent Control and Robotics, School of Automation, Hangzhou Dianzi University, Hangzhou 310018, China; lihan@hdu.edu.cn (H.L.); lzycris@hdu.edu.cn (Z.L.); frgao@hdu.edu.cn (F.G.)

**Keywords:** brain network, tensor decomposition, epilepsy prediction

## Abstract

Epilepsy is a chronic neurological disorder which can affect 65 million patients worldwide. Recently, network based analyses have been of great help in the investigation of seizures. Now graph theory is commonly applied to analyze functional brain networks, but functional brain networks are dynamic. Methods based on graph theory find it difficult to reflect the dynamic changes of functional brain network. In this paper, an approach to extracting features from brain functional networks is presented. Dynamic functional brain networks can be obtained by stacking multiple functional brain networks on the time axis. Then, a tensor decomposition method is used to extract features, and an ELM classifier is introduced to complete epilepsy prediction. In the prediction of epilepsy, the accuracy and F1 score of the feature extracted by tensor decomposition are higher than the degree and clustering coefficient. The features extracted from the dynamic functional brain network by tensor decomposition show better and more comprehensive performance than degree and clustering coefficient in epilepsy prediction.

## 1. Introduction

Epilepsy is a kind of neurological disorder characterized by recurrent epileptic seizures which can affect 65 million patients worldwide [1]. During seizures, abnormal discharges of neurons in the brain can lead to various symptoms [2]. The sudden occurrence of seizures may endanger the patient’s life, as patients may lose control of their bodies when performing complex tasks. Seizure prediction can help to alert patients and avoid some injuries. Recording brain activity is usually based on invasive techniques, such as electrocorticography (ECoG) and non-invasive techniques, such as electroencephalography (EEG). The ECoG signal has low susceptibility to artifacts and high signal-to-noise ratio. EEG is non-invasive and low cost, and can be used for long-term monitoring [3,4,5]. EEG is of great significance in the study of epilepsy, and has been proved to be effective in studying brain function [3,4].

Based on the special characteristics of EEG, many researchers have applied different techniques to investigate epileptic seizures prediction. Parvez and Paul [6] used phase correlation to calculate spatiotemporal correlation among EEG signals, and applied this method to seizure prediction. Vidyaratne et al. [7] applied a method based on wavelet decomposition and fractal dimension to detect epileptic seizure from scalp EEG. EEG is usually divided into five basic waves based on frequency: delta, theta, alpha, beta and gamma [8]. Ghosh-Dastidar et al. [9], Subasi et al. [10] and Zhang et al. [11] divided EEG into several frequency bands to extract the features of EEG signals. Parvez et al. [12] claimed that the features extracted from gamma frequency should provide better classification accuracy between seizure ictal and interictal. Zhang et al. [11] obtained the best epilepsy prediction accuracy in the beta band. Therefore, it is feasible to divide EEG into different frequency bands.

Recently, the brain is increasingly regarded as a dynamic complex network [13]. Network based analyses is of great help in the investigation of seizures, because they show that the global and regional characteristics of the human brain change during seizures [14,15]. As the study of a single feature in EEG is considered insufficient, functional brain networks are developing rapidly in network analysis and have become more and more important in many neurological studies [16,17]. Driscoll et al. [18] claimed that epilepsy could be caused by brain network disorders. Now graph theory is commonly applied to analyze functional brain networks [19]. Mitsis et al. [20] built brain networks through cross-correlation, coherence and corrected cross-correlation, and investigated the correlation between functional brain network and seizure onset. Yu et al. [21] reconstructed the brain network by phase synchronization index matrices, and calculated local efficiency and globe efficiency to investigate the alternation in the brain network structure of patients with epilepsy. Zhang et al. [11] used nonlinear partially directed coherence as a measure of functional brain networks and applied this to seizure detection. Based on this, an average accuracy of 84.4% and an average prediction time of 1265.43 s could be achieved.

At present, many studies apply graph theory to analyze brain functional networks in a resting state [19,22], but functional brain networks are dynamic [23,24,25], and the onset of epilepsy will lead to changes in them [26,27]. Fallahi et al. [22] applied a dynamic functional brain network approach to identify temporary lobe epilepsy, and obtained higher accuracy than with static features. Therefore, it is feasible to apply dynamic functional brain network for epilepsy prediction. Stacking multiple functional brain networks into a high-dimensional array with time as the axis, this array can be regarded as a dynamic functional brain network. In the field of mathematics, the higher-order matrix can be called ‘tensor’ [28], and tensor decomposition can extract features from the tensor.

Tensor decomposition has found wide applications in areas such as feature extraction and biomedical signal processing [28]. In recent years, tensor decomposition is a good method to extract the main characteristics of data [29]. Thanh et al. [30] developed an automatic detection system for epileptic spikes using simultaneous multilinear low-rank approximation of tensors. Spyrou et al. [29] used Tucker decomposition to extract valuable features from epileptic data and applied machine learning and tensor decomposition to construct a framework for analyzing epilepsy EEG data. Now there are two widely used tensor decomposition models: CP (CANDECOMP/PARAFAC) decomposition and Tucker decomposition [28]. Both models can extract the main features of data.

In this study, we attempted to develop a new method of epilepsy prediction based on tensor decomposition. Specifically, the EEG was divided into small segments by a sliding window, and then each time segment was constructed as a brain functional network. Afterwards, several brain functional networks were stacked into dynamic brain functional networks (tensors) with time as the axis. CP decomposition was used to extract features from the obtained tensor to develop an epilepsy prediction method based on the extreme learning machine (ELM).

## 2. Materials and Methods

In this study, EEG was first preprocessed. For the preprocessed EEG, the correlation coefficient was used to calculate the functional brain networks. The dynamic functional brain networks could be obtained by stacking the functional brain network on the time axis, and then tensor decomposition was applied to extract features. The features extracted by tensor decomposition and ELM were used for epilepsy prediction. Figure 1 shows the main flow chart.

### 2.1. Database

In this paper, the public database provided by MIT (CHB-MIT) [31] is adopted. The data were collected from 22 subjects. All data were sampled at 256 Hz. The international standard 10–20 electrode distribution system was used to record the signal (Figure 2). Most cases contain 23 channels, and a few cases contain less or more channels. Eighty-eight epileptic seizure data sets were obtained from 11 patients. Each EEG data set was intercepted over different time periods, obtaining 44 inter-ictal EEG segments and 44 pre-ictal EEG segments. The pre-ictal EEG segments are 45 min before the seizure, and 44 similar 45-min EEG segments during the inter-ictal period have no overlap with the pre-seizure segment [32].

### 2.2. Data Preprocessing

Then, a filter based on Fourier transform was applied to divide the EEG into five bands. These five bands–gamma (30–60 Hz), beta (15–30 Hz), alpha (8–15 Hz), theta (4–8 Hz) and delta (0.5–4 Hz)–and unfiltered raw EEG were used for subsequent calculations.

### 2.3. Functional Brain Network Construction

To create a functional network, a matrix containing the EEG channels’ pairwise correlations is required. Therefore, Pearson product-moment correlation coefficient, which was first developed by Pearson in 1895 [33], was applied to calculate the correlation matrix, where the rows and columns both represent channels. The value of the correlation matrix represents the correlation between channels [34].

In this paper, an 8-s time window was used to segment the EEG data, and the time window used 4-s steps to slide on the data [11]. Then, correlation coefficient was applied to calculate the correlation matrix of each 8-s data segment.

To binarize the correlation matrix, a thresholding was applied to identify strongly correlated nodes. A statistical significance threshold (Th) [35] was adopted:(1)$$Th=1-{\left(1-\alpha \right)}^{1/(L-1)}$$ where L is the average number of disjoint sections mentioned in the above procedure, and α is the required confidence level with a value of 0.95 in this paper [11]. If the correlation coefficient was greater than thresholding, the two channels discussed were considered to be correlated. The value of these directional relationships was 1. On the other hand, when the correlation coefficient was less than thresholding, the two channels were considered to be independent of each other, and they were assigned a value of 0. Since 23-channel EEG signals were used, the functional brain networks obtained were 23×23 matrices.

### 2.4. Degree and Clustering Coefficient

In the present study, clustering coefficient and degree were used to analyze the functional brain network, in order to compare with the features extracted by tensor decomposition. Degree and clustering coefficient have been widely used and have shown their advantages in classifying epileptic states [11,36].

The degree of a node represents the number of neighbor nodes owned by the node. The degree of node i is defined as:(2)$$D_{i}=\sum_{j=1}^{J}a_{ij}$$
where J is the number of nodes, and $a_{ij}$ is the element of the functional brain networks. If $a_{ij}$ is 0, node i and node j are not connected, and if $a_{ij}$ is 1, node i and node j are connected.

The clustering coefficient can describe the degree of tight connection between nodes in the network [13]. For a complex network, the clustering coefficient of nodes is defined as the actual number of edges in adjacent nodes divided by the maximum number of edges that may exist in adjacent nodes.
(3)$$C_{i}=\frac{2E_{i}}{k_{i}(k_{i}-1)}$$
where $E_{i}$ is the actual number of edges in adjacent nodes of node, and $k_{i}$ is the number of adjacent nodes.

### 2.5. Tensor Construction

A tensor is a multidimensional matrix. More formally, one element of the *n* vector space tensor product is an *n*th-order tensor. The number of dimensions is the order of a tensor [28]. In this paper, after calculating the functional brain networks, the dynamic functional brain network can be obtained by stacking multiple functional brain networks on the time axis. Since the functional brain networks were 23×23 matrices, the dynamic brain functional network constructed in this way can be regarded as a third-order tensor $T\in R^{23\times 23\times num}$. In order to analyze the effect of num value on the prediction of epilepsy, num has four values, 3, 4, 5 and 6 in this paper. The schematic diagram of tensor construction is shown in Figure 3.

### 2.6. Tensor Decomposition

Tensor decomposition is an extension of matrix analysis. It has found wide applications in areas such as feature extraction and biomedical signal processing [28]. Tensor is the representation of high-dimensional data, and the model is often more complex. Tensor decomposition is a good method to extract the main features of data [29]. In recent years, tensor decomposition has received more attention from many researchers in various fields.

There are two well-known kinds of high order tensor decomposition: CP decomposition and tucker decomposition. CP decomposition is also known as CANDECOMP/ PARAFAC decomposition, which can decompose tensors into the sum of rank-one tensors [28]. CP decomposition was mainly used in this study.

A rank-one tensor is defined as a tensor which can be expressed as the outer product of N vectors in the following way,
(4)$$X=a^{\left(1\right)}\circ a^{\left(2\right)}\circ \cdots \circ a^{\left(N\right)}$$

The symbol ∘ is the outer product.

For CP decomposition, it is expected that a third-order tensor $X\in R^{I\times J\times K}$ can be represented as follows,
(5)$$X\approx \sum_{r=1}^{R}a_{r}\circ b_{r}\circ c_{r}$$
where R is the number of components when CP decomposition is calculated, $a_{r}\;\in \;R^{I}$, $b_{r}\;\in \;R^{J}$, and $c_{r}\;\in \;R^{K}$ for r = 1, ..., R [20].

If the columns of the factor matrices are normalized and the weights are absorbed into the vector λ
(6)$$X\approx \sum_{r=1}^{R}\lambda_{r}a_{r}\circ b_{r}\circ c_{r}$$

If R is small, the tensor decomposition will have a large error. On the other hand, a large R will lead to a long decomposition time. In this paper, we choose the value of R by error. Given a tensor, the initial value of R is set to 1. Then carry out CP decomposition and obtain factor matrices. Calculate the error between the tensor constructed by the factor matrices and the given tensor. The initial value of R will continue to increase until the error is less than 0.05. The flow chart is shown in Figure 4.

Now there are many methods to calculate CP decomposition [28]. In this study, the alternating least squares (ALS) method was adopted, which was proposed by Carroll and Carroll [37] and Harshman [38].

### 2.7. Feature Extraction

Functional brain networks can reflect the activities of the brain [21]. After calculating the functional brain networks, they were stacked to form tensors according to the method of Section 2.5. In this study, tensor decomposition is adopted to extract features from the obtained tensors.

In CP decomposition, different factor matrices can be obtained by decomposing different tensors, which is not conducive to the analysis of decomposition results and the extraction of tensor features. In this study, when the tensor was decomposed for the first time, the factor matrix A of channel×R, the factor matrix B of channel×R, the factor matrix C of time×R, and the coefficient vector λ of 1×R could be obtained. The matrices A and B were retained. In the next decomposition, the matrices A and B were fixed, and C and λ were calculated to extract features. The matrix F was obtained by multiplying C and λ by columns, and the column average of F was extracted as a feature.

The pseudo-code of extract features using tensor decomposition is shown in Algorithm 1.
**Algorithm 1: Extract Features Using Tensor Decomposition****Input:** EEG signals.**Output:** features extracted by tensor decomposition.  1: Preprocess EEG signal.  2: Construct functional brain networks.  3: Construct tensors by stacking multiple functional brain networks on the time axis.  4: **if** Matrices A and B are not fixed **then**  5: Estimating R.  6: Perform tensor decomposition and result in {matrices A, B, C and vector λ}.  7: Fix matrices A and B.  8: Calculate features by matrix A and vector λ.  9: **else if** Matrices A and B are fixed **then**10: Perform tensor decomposition and result in {matrix C and vector λ}.11: Calculate features by matrix A and vector λ.12: **end if**.13: **return** Features extracted by tensor decomposition.

### 2.8. Classifier

In order to predict epileptic seizures, ELM was selected to classify epileptic features extracted by tensor decomposition.

ELM is a feedforward neural network which is much faster in training. It can generate input weights randomly and directly find the optimal solution of the output weight matrix without iterative optimization [39].

For training data ${\left\{(x_{i},y_{i})\right\}}_{i=1}^{N}\in R^{d}\times R^{l}$, where $x_{i}={\left[x_{i1},x_{i2},\cdots ,x_{id}\right]}^{T}$ is the input feature vector, and $y_{i}={\left[y_{i1},y_{i2},\cdots ,y_{il}\right]}^{T}$ is the output vector.

Assuming the ELM has M hidden neurons, the estimated value of the hidden layer output target vector is
(7)$${\hat{y}}_{i}=\sum_{j=1}^{M}\beta_{j}g(w_{j}^{T}\cdot x_{i}+b_{j})\;\;\;i=1,2,\cdots N$$
where $w_{j}={\left[w_{j1},w_{j2},\cdots ,w_{jd}\right]}^{T}$ is the input weight vector, $b_{i}$ is the bias of the hidden neuron, g(•) is the activation function, and $\beta_{j}={\left[\beta_{j1},\beta_{j2},\cdots ,\beta_{jl}\right]}^{T}$ is the output weight vector.

Equation (7) can be transformed into the matrix form
(8)$$\hat{Y}=H\beta \;$$
where $\beta ={\left[\beta_{1}\cdots \beta_{m}\right]}^{T}$ is the output weight matrix, H is the hidden-layer output matrix, and $\hat{Y}={\left[{\hat{y}}_{1}\cdots {\hat{y}}_{N}\right]}^{T}$ is the output vector.
(9)$$H={\left[\begin{array}{ccc} g(w_{1}^{T}x_{1}+b_{1}) & \cdots & g(w_{M}^{T}x_{1}+b_{M})\\ \vdots & \ddots & \vdots \\ g(w_{1}^{T}x_{N}+b_{1}) & \cdots & g(w_{M}^{T}x_{N}+b_{M}) \end{array}\right]}_{N\times M}$$

For the ELM model, the optimization of the model can be achieved by setting the weight vector $\;w_{j}\;$ and bias $b_{i}$ randomly. The core of ELM is to get the weight matrix $\overset{⌢}{\beta }$ such that
(10)$$\overset{⌢}{\beta }=\underset{\beta }{\arg\min}\|H\beta -\overset{⌢}{Y}\|$$

This can be solved by the least square method as shown in Equation (11)
(11)$$\hat{\beta }=H^{+}\hat{Y}$$

Compared with the traditional feedforward neural network, ELM just needs one step to calculate the output weight. In this way, ELM improves the learning speed and generalization ability, and reduces the time cost of the network.

Regularization is commonly used to optimize ELM:(12)$$\overset{⌢}{\beta }=\underset{\beta }{\arg\min}\|H\beta -\overset{⌢}{Y}\;\|+\frac{1}{E}\|\beta \|$$
where E is the regularization parameter. The weight matrix $\overset{⌢}{\beta }$ is:(13)$$\overset{⌢}{\beta }=\{\begin{array}{c} {\left(H^{T}H+EI\right)}^{-1}H^{T}\overset{⌢}{Y}\;\;\;M\leq N\\ H^{T}{\left(HH^{T}+EI\right)}^{-1}\overset{⌢}{Y}\;\;\;M>N \end{array}$$
where I denotes the identity matrix. In this study, the leave-one-out cross validation approach proposed by Cao et al. [40] is applied to optimize the regularization parameters of ELM.

### 2.9. Seizure Onset Prediction

In this part, a strategy of epilepsy prediction is proposed. Its flowchart is shown in Figure 5.

For every 45-min EEG segment, the EEG data was segmented by a semi overlapping 8-s sliding window. The functional brain networks which were constructed by segmented EEG signals were used to construct tensors. Epileptic features were extracted from these tensors by tensor decomposition. Since many data fragments were segmented during the pre-ictal and inter-ictal periods, the "leave-one-out" method was used to conduct epilepsy prediction experiments. More specifically, one 45-min data fragment was applied as the testing set and the rest were used to train the ELM classifier [41]. Then, the trained classifier was used to continuously identify the state of the testing set, and the labels of inter-ictal and pre-ictal were designated as 1 and 2, respectively. Set the initial alarm to 0. When the pre-ictal state is detected, i.e., the label is detected as 2, alarm plus one. Otherwise alarm is set to zero. For each EEG segment, if 30 consecutive pre-ictal states can be detected, i.e. alarm = 30, epilepsy alert will be triggered. For each pre-ictal EEG segment, if an alert was triggered before the onset of the seizure, the prediction was considered to be correct. For each inter-ictal EEG segment, if an alert failed to be triggered, the prediction was considered to be correct. The prediction time of a pre-ictal data fragment was:(14)T=s×(n+1)
where n represents the number of sliding windows which is not processed before epilepsy is predicted, and s is the step size (4 s in this paper). For each frequency band, the prediction time was the average time required to successfully predict epilepsy.

## 3. Results

In this paper, unfiltered full band signals and 5 bands signals (delta, theta, alpha, beta and gamma) were used to construct a brain functional network. Then four num values were applied to construct a dynamic brain functional network. Features could be extracted by tensor decomposition. Before tensor decomposition, the parameter R was selected by error. The curve between R and error is shown in Figure 6. The time consumed in tensor decomposition of dynamic functional brain networks with different num values is shown in Figure 7. When the values of num are 3, 4, 5 and 6 respectively, the values of R are 27, 34, 38 and 43.

In order to compare the performance differences between the features extracted by tensor decomposition and degree and clustering coefficient, ten-fold cross validation was used to classify and test the features of each band. The results of degree and clustering coefficient are shown in Table 1, and the results of features extracted by tensor decomposition are shown in Table 2. It can be seen that the features extracted by tensor decomposition has higher accuracy, specificity, sensitivity and f1 score than degree and clustering coefficient. It can be seen that the feature extracted by tensor decomposition has a more significant difference in classification than by degree and clustering coefficient.

Then the features extracted by tensor decomposition and degree and clustering coefficient were applied to predict epileptic seizures. The prediction results of degree and clustering coefficient are shown in Table 3, and the prediction results of features extracted by tensor decomposition are shown in Table 4.

## 4. Discussion

In this paper, functional brain networks are constructed by Pearson correlation coefficient, and dynamic functional brain networks can be obtained by stacking multiple functional brain networks on the time axis. After that, the features can be extracted by the tensor decomposition and then complete epilepsy prediction can occur with ELM. The results demonstrated that the features which were extracted from the dynamic functional brain networks by tensor decomposition show better comprehensive performance in epilepsy prediction than degree and clustering coefficient.

Table 3 shows the results of epilepsy prediction using degree and clustering coefficient. Table 4 shows the results of epilepsy prediction using the features extracted by tensor decomposition. From these two tables, we can see that, in any frequency band, epilepsy prediction with degree and clustering coefficient will arrive at higher specificity, but the sensitivity is relatively poor. Applying the features extracted by tensor decomposition to predict epilepsy will arrive at higher sensitivity, but the specificity is lower than the degree and clustering coefficient. In addition, the accuracy, F1 score and prediction time of the feature extracted by tensor decomposition are higher than the degree and clustering coefficient. Compared with degree, when num is 6, the accuracy of epilepsy prediction using the features extracted by tensor decomposition is 7.48% higher on average. Compared with clustering coefficient, when num is 6, the accuracy of epilepsy prediction using the features extracted by tensor decomposition is 12.73% higher on average. In order to compare the effect of epilepsy prediction, the *p*-value between the accuracy of epilepsy prediction using features extracted by tensor decomposition and the accuracy of epilepsy prediction using degree and clustering coefficient was calculated. When comparing with degrees, the full-band with num of 3 has the maximum *p*-value of 0.0006, and the *p*-values of other bands with other num values are less than this value. When compared with the clustering coefficient, the full-band with num of 5 has the maximum *p*-value of 0.00002, and the *p*-values of other bands with other num values are less than this value. This shows that the features which were extracted by tensor decomposition have higher comprehensive performance in epilepsy prediction compared to the degrees and clustering coefficient, and using tensor decomposition to extract features for epilepsy prediction has statistical significance regarding improvement of the prediction effect.

In Table 4, the results of epilepsy prediction using the features extracted by tensor decomposition in each band are shown. Among these bands, the gamma band has the highest F1 score, sensitivity and accuracy, but the specificity is lower than in some bands. Other bands also have good prediction results, which can be applied to predict seizures. These results highlight recent findings. Wang et al. [13] found that epilepsy can cause functional brain network changes of different frequencies. In addition, the full band signal without filtering is used in the epilepsy prediction experiment, and the effect is second only to the gamma band. If there is a need to reduce the amount of computation and computing time, the use of unfiltered full band signal is also an option.

In Table 4, the results of extracting features from dynamic brain functional networks constructed from four num values using tensor decomposition and making epilepsy predictions are demonstrated. Among these, a num of 3 gives the best specificity in predicting seizure, but the lowest sensitivity. As the num value increases, the specificity decreases while the sensitivity rises gradually. Among the four num values used in this paper, the F1 score and accuracy rate are the highest when num is 6, which illustrates that the optimal comprehensive performance can be achieved here, but the increase of num does not absolutely lead to the increase of comprehensive performance. There are partial bands at num = 4 with lower F1 score and accuracy than at num = 3. Figure 6 shows that as the num value increases, the parameter R also increases when carrying out tensor decomposition to ensure that the error is within a certain range. Figure 7 shows that the time to perform tensor decomposition also increases. Therefore, the num should not be too large or too small, and factors such as specificity, sensitivity, and time of calculation need to be considered in combination.

There were some limitations in this study. In the process of extracting features from dynamic brain functional networks using tensor decomposition, partial factor matrices were fixed. These factor matrices were generated when the tensor was decomposed, so they have some randomness. In this study, the influence of fixed factor matrices on the prediction of epilepsy was not excluded. In this direction, further experiments can be performed to determine whether fixed matrices have impact on epilepsy prediction, as well as to optimize. Furthermore, in this paper Pearson’s correlation was used as coefficient when constructing brain functional networks and ELM was applied when predicting seizure, but these methods may not be the optimal ones. Methods with higher performance can be found in the following work. Above are our future work plans.

## 5. Conclusions

In this study, correlation coefficient is applied to create a functional brain network, and dynamic functional brain networks can be obtained by stacking multiple functional brain networks on the time axis. Then, a tensor decomposition method is used to extract features, and an ELM classifier is introduced to complete epilepsy prediction. The test shows that the features extracted from the dynamic functional brain network by tensor decomposition have better comprehensive performance than degree and clustering coefficient in epilepsy prediction. This study provides a new method to extract features from the brain functional network.

## Figures and Tables

**Figure 1 brainsci-11-01066-f001:**
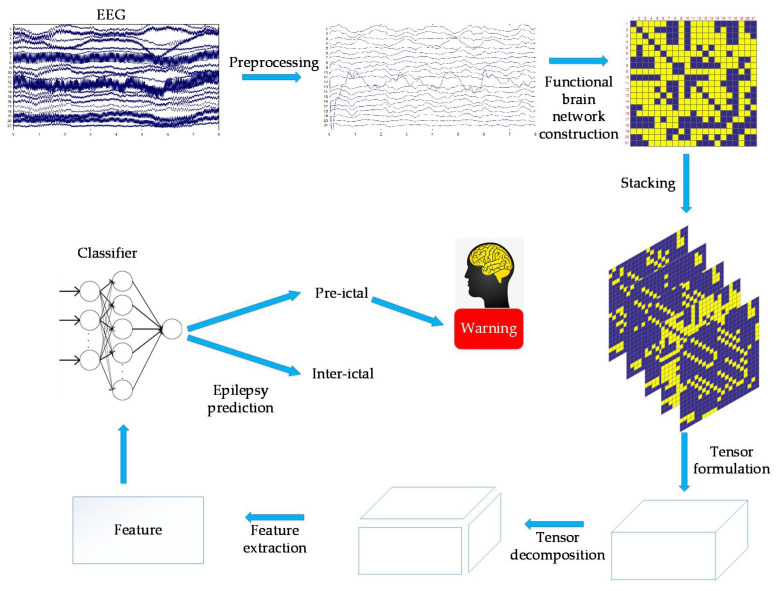
This figure shows the main flow chart.

**Figure 2 brainsci-11-01066-f002:**
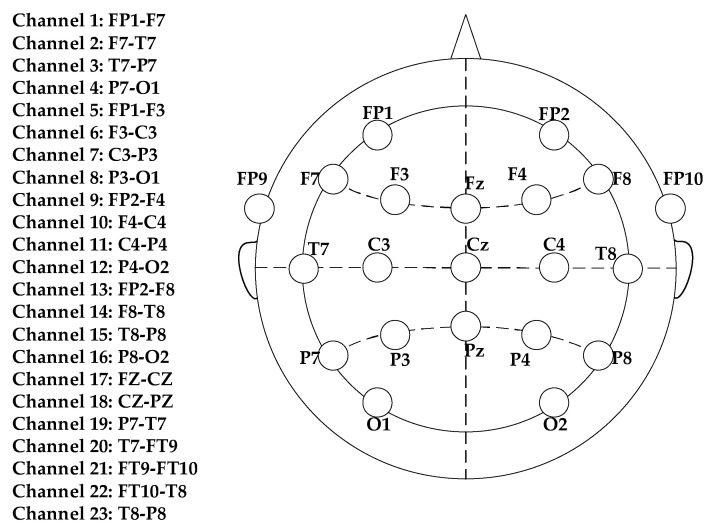
This figure shows the placement of the EEG electrodes.

**Figure 3 brainsci-11-01066-f003:**
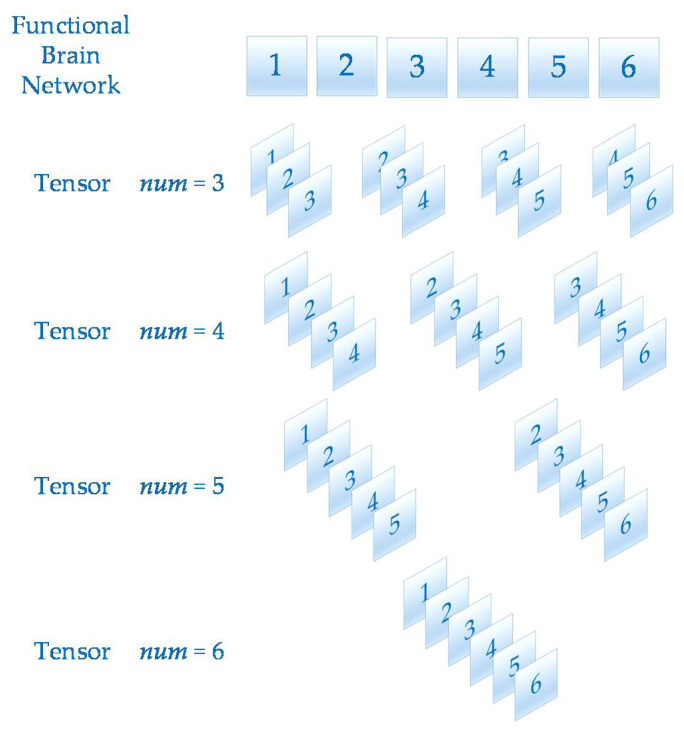
This figure shows the tensor construction of different num values.

**Figure 4 brainsci-11-01066-f004:**
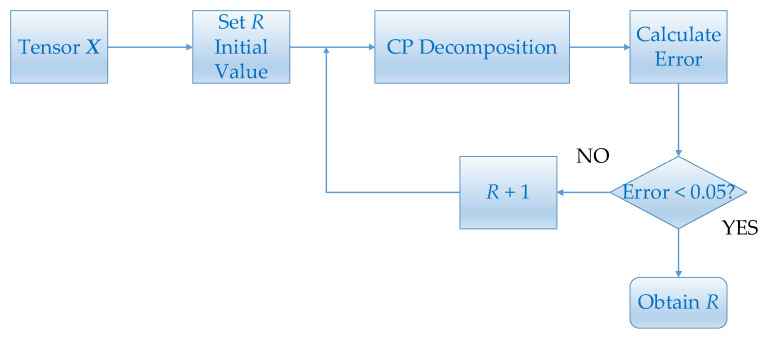
This figure shows the flowchart of estimating R.

**Figure 5 brainsci-11-01066-f005:**
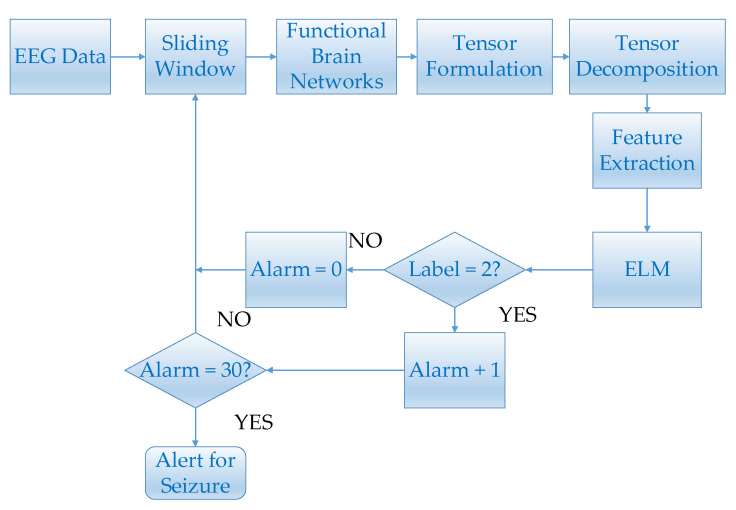
This figure shows the flowchart of epilepsy prediction.

**Figure 6 brainsci-11-01066-f006:**
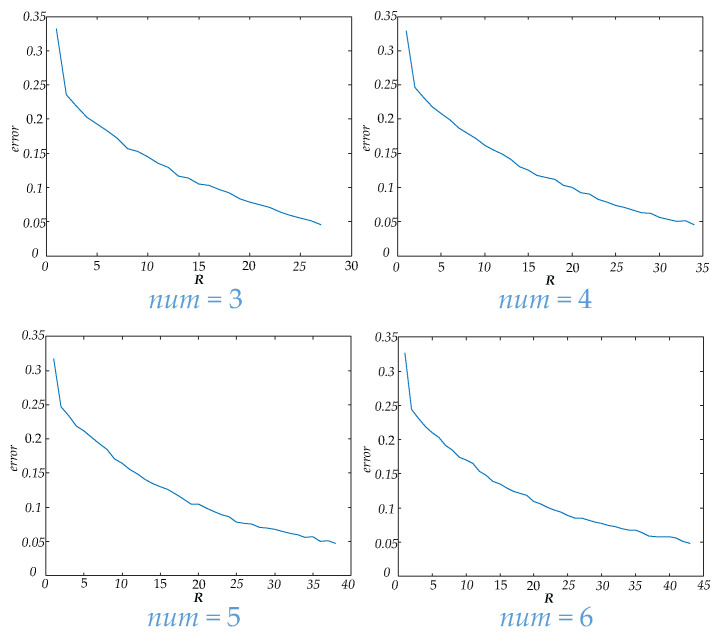
This figure shows the curves of R and error for different num values.

**Figure 7 brainsci-11-01066-f007:**
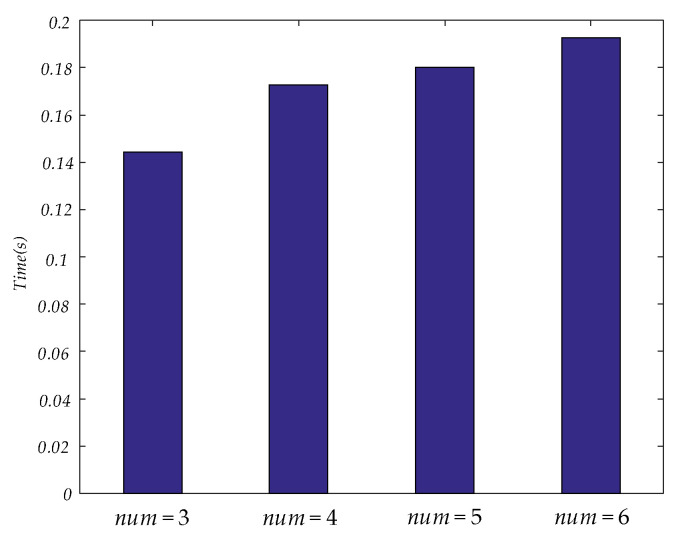
This figure shows the time consumed in tensor decomposition of dynamic functional brain networks with different num values.

**Table 1 brainsci-11-01066-t001:** This table shows the classification results of degree and clustering coefficient.

		Full-Band	Delta	Theta	Alpha	Beta	Gamma
Degree	F1 Score	0.82	0.77	0.77	0.79	0.83	0.85
	Specificity (%)	84.52	78.82	79.23	81.60	83.80	83.97
	Sensitivity (%)	80.94	75.67	75.54	76.78	82.44	85.56
	Accuracy (%)	82.73	77.24	77.38	79.21	83.12	84.77
Clustering coefficient	F1 Score	0.79	0.72	0.73	0.75	0.78	0.81
	Specificity (%)	82.36	75.26	75.94	77.91	77.45	79.96
	Sensitivity (%)	77.47	70.54	71.88	73.23	78.12	81.35
	Accuracy (%)	79.93	72.92	73.91	75.56	77.80	80.66

**Table 2 brainsci-11-01066-t002:** This table shows the classification results of the features extracted by tensor decomposition.

		Full-Band	Delta	Theta	Alpha	Beta	Gamma
num = 3	F1 Score	0.86	0.82	0.82	0.83	0.87	0.88
	Specificity (%)	87.31	83.49	84.09	85.28	87.18	87.34
	Sensitivity (%)	85.26	81.42	81.29	80.81	86.56	88.64
	Accuracy (%)	86.30	82.47	82.68	83.07	86.89	88.00
num = 4	F1 Score	0.86	0.81	0.82	0.82	0.87	0.88
	Specificity (%)	87.03	83.08	82.82	86.10	87.66	87.89
	Sensitivity (%)	84.57	80.58	80.79	80.23	87.04	88.08
	Accuracy (%)	85.81	81.81	81.81	83.21	87.36	87.99
num = 5	F1 Score	0.87	0.83	0.82	0.84	0.88	0.90
	Specificity (%)	87.53	84.01	83.46	86.36	88.81	88.92
	Sensitivity (%)	85.81	81.67	81.63	82.72	87.64	89.86
	Accuracy (%)	86.67	82.86	82.54	84.58	88.24	89.40
num = 6	F1 Score	0.89	0.85	0.85	0.87	0.90	0.90
	Specificity (%)	89.18	85.96	85.87	88.80	90.25	89.84
	Sensitivity (%)	88.03	84.60	84.42	84.88	89.46	89.91
	Accuracy (%)	88.62	85.28	85.15	86.86	89.87	89.88

**Table 3 brainsci-11-01066-t003:** This table shows the prediction results of degree and clustering coefficient.

		Full-Band	Delta	Theta	Alpha	Beta	Gamma
Degree	F1 Score	0.68	0.56	0.57	0.60	0.69	0.70
	Specificity (%)	88.00	89.33	95.56	89.78	81.78	88.44
	Sensitivity (%)	58.18	43.64	41.82	47.27	61.82	60.45
	Accuracy (%)	73.26	66.74	68.99	68.76	71.92	74.61
	Prediction time (s)	2013.68	1851.21	1914.39	1866.15	2048.42	1980.42
Clustering coefficient	F1 Score	0.62	0.41	0.49	0.44	0.54	0.69
	Specificity (%)	94.22	95.11	97.33	91.11	76.44	81.33
	Sensitivity (%)	47.27	26.82	33.18	30.45	46.36	62.73
	Accuracy (%)	71.01	61.35	65.61	61.12	61.57	72.13
	Prediction time (s)	1823.15	1731.66	1634.79	1553.85	1906.71	2004.17

**Table 4 brainsci-11-01066-t004:** This table shows the prediction results of the features extracted by tensor decomposition.

		Full-Band	Delta	Theta	Alpha	Beta	Gamma
num = 3	F1 Score	0.76	0.70	0.74	0.70	0.78	0.80
	Specificity (%)	84.44	80.44	85.78	85.33	78.22	75.56
	Sensitivity (%)	70.91	65.00	66.82	61.82	77.73	83.18
	Accuracy (%)	77.75	72.81	76.40	73.71	77.98	79.33
	Prediction time (s)	2077.64	1858.49	1983.05	2087.18	2070.76	2072
num = 4	F1 Score	0.75	0.70	0.71	0.72	0.79	0.79
	Specificity (%)	80.00	74.22	80.00	76.44	74.22	75.56
	Sensitivity (%)	71.82	68.18	66.82	69.09	83.18	81.82
	Accuracy (%)	75.96	71.24	73.48	72.81	78.65	78.65
	Prediction time(s)	2151.54	1976.69	2066.42	2035.39	2109.72	2191.64
num = 5	F1 Score	0.79	0.74	0.78	0.73	0.80	0.80
	Specificity (%)	76.89	71.56	76.89	75.56	72.89	73.33
	Sensitivity (%)	80.91	75.45	79.55	71.36	85.00	85.00
	Accuracy (%)	78.88	73.48	78.20	73.48	78.88	79.10
	Prediction time (s)	2055.50	1868	1893.39	1929.10	2058.25	2144.47
num = 6	F1 Score	0.80	0.76	0.79	0.76	0.80	0.82
	Specificity (%)	74.22	70.67	75.56	76.00	70.67	73.33
	Sensitivity (%)	85.00	80.45	81.82	76.82	86.36	88.18
	Accuracy (%)	79.55	75.51	78.65	76.40	78.43	80.67
	Prediction time (s)	2057.77	1995.41	2060.24	2031.41	2132.65	2117.23

## Data Availability

CHB-MIT EEG Database. https://physionet.org/content/chbmit/1.0.0/, accessed on 21 May 2021.

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
