# Peer review of "Prediction of Epilepsy Based on Tensor Decomposition and Functional Brain Network"

_brainsci, 2021, doi:10.3390/brainsci11081066_

Round 1

Reviewer 1 Report

Summary of the manuscript

Li et al characterizes EEG signals recorded from patients with epilepsy. By using graph theoretical approach and tensor decomposition, the authors demonstrate that EEG-based features extracted through tensor decomposition are highly predictable of seizure onset. Further, these ‘biomarkers’ outperform graph-theoretical features such as network degree and clustering coefficient in seizure prediction.

General assessment
The strengths of this manuscript are two-fold. First, the authors devised a well-thought-out analytical framework that can be used to capture state changes in neural recordings (EEG) in epilepsy. Second, the topic discussed is well-motivated and has important implications for clinical practice as well as basic science research investigating the neural mechanisms underlying epileptic seizures.

Having said this, I found certain information and details concerning the analytical technique to be missing. Moreover, while the schematics of the analysis pipeline are well presented, figures (and stats) that illustrate the main results of prediction accuracy should be included in addition to the tables.

Major comments

  1. Could the authors discuss why only frequency-specific activity, but not broad-band signal was considered for the analyses? Are there specific hypotheses as to why seizure predictability would be best achieved through frequency-specific activity?
  2. I believe that the discussion of the main findings could benefit from explicit discussion/speculation as to why tensor decomposition is a better method than graph theory in characterizing seizure states. To do so, I recommend citing recent findings that have used various graph-theoretical features in capturing different aspects of seizures. Some samples of such studies with high relevance to the present findings are:
  • Bomela et al., Scientific Reports (2020). Real-time inference and detection of disruptive EEG networks for epileptic seizures.
  • Rungratsameetaweemana et al., bioRxiv (2021). Brain network dynamics codify heterogeneity in seizure propagation.
  • Lariviere et al., bioRxiv (2021). Network-based atrophy modelling in the common epilepsies: a worldwide ENIGMA study.
  • Driscoll et al., Communications Biology (2021). Multimodal in vivo recording using transparent graphene microelectrodes illuminates spatiotemporal seizure dynamics at the microscale.
  1. While the prediction results of each method are presented in tables, figures with associated statistical results are needed to clearly illustrate the findings. Including statistical differences between predictability associated with each method is especially important in the present manuscript (and in the context of epilepsy research in general) due to the high inter-subject variability and the small size of the data used.
  2. Out of curiosity, do the authors suspect the patterns of findings to hold if ECoG data were used instead of EEG? Also, can the authors comment on why EEG signals were used in the present study when ECoG recordings are often available in patients with epilepsy?

Reviewer 2 Report

The manuscript in the present form is inaccessible to the Brain Sciences readers. The network tensor-based method for seizure prediction is not sufficiently described for each step of the calculations. For example, what is R? why does it have dimensions 23x23xnum? It would be best if the authors include a computer code for one example and describe the step-by-step calculations. In what way and by how much, this method is better than the graph theoretic measures? 

Round 2

Reviewer 1 Report

The authors have carefully addressed my concerns raised in the first round. I believe the manuscript is ready for publication after minor language check.

Reviewer 2 Report

The authors have appropriately addressed everything that was in the review report and  the revised manuscript looks acceptable for publication.